# Validity and Applicability of the Scaling Effects for Low Velocity Impact on Composite Plates

**DOI:** 10.3390/ma14195884

**Published:** 2021-10-08

**Authors:** Michele Guida

**Affiliations:** Department of Industrial Engineering, Università degli Studi di Napoli Federico II, 80125 Napoli, Italy; michele.guida@unina.it

**Keywords:** scaling, similitude, low velocity impact, composite, carbon fiber

## Abstract

As a result of the increasing use of composite materials in engineering fields, the study of the effect of scale on impact performance is essential for the design of large-scale structures. The purpose of this study was to develop a method capable of identifying a corrective factor that can be used to evaluate based on similarity theory the behavior of panels with the same material but with scaled geometry when subjected to low velocity impact. The field of investigation was applied based on the experimental results present in the bibliography and that refer to two flat sheets differing only in geometric scale and made by overlapping carbon/carbon unidirectional pre-impregnated epoxy 914 C-TS (6K) −5 34% sheets. Behavior outside the range of structural linearity was investigated for the scaled panels, and the theoretical predictions of the model, projected with each law of scale for each variable present in the dynamic impact process, were compared with the experimental data. A finite element model was thereby developed that validates the theory of scaling and its limits of applicability up to the limits of fracture.

## 1. Introduction

The spread of composite materials in the realization of transport structures has played a significant role. The enormous design flexibility of advanced composites is obtained at the cost of large number of design parameters. Furthermore, any new design is extensively evaluated experimentally until it achieves the necessary reliability, performance and safety. However, the experimental evaluation of composite structures is costly and time consuming. Consequently, it is extremely useful if a full-scale structure can be replaced by a similar scaled-down model that is much easier to work with. Furthermore, a dramatic reduction in cost and time can be achieved if available experimental data of a specific structure can be used to predict the behavior of a group of similar system.

The design flexibility of advanced composites comes at the cost of an extensive number of parameters, and the need for material characterization and subsequent experimental evaluation guarantees an increase not only in terms of reliability but also in terms of safety and performance. Researchers focus on the need for scale models that can replace full-scale structures. Thus, if the available experimental data of a scale structure can be replicated on a real scale, a drastic reduction in costs and time can be obtained.

Similarity theory defines the laws of scale, which form a relationship between a full-scale structure and its scale model and can be used to extrapolate the experimental data of a small, inexpensive and verifiable model into design information for a large prototype. There are two methods to develop similarity conditions, the direct use of governance equations and dimensional analysis. The similarity conditions can be established either directly from the field equations of the system or, if it is a new phenomenon and the mathematical model of the system is not available, by means of dimensional analysis. The first method is more convenient than dimensional analysis since the resulting similarity conditions are more specific. In this case, the behavior of the system in terms of variables and parameters are characterized by the field equations of the system with their own boundary and initial conditions. Simitses focused on the direct use of governance Equations (1) and (2). In his studies he only evaluates the direct use of the government equation procedure; the two scaled and real systems are similar if the field equations are similar.

This similarity defines the scaling laws of the two systems starting from the structural geometry and consequently between cause and response of the two systems. In this approach, it is necessary to make sure that all the variables and parameters that influence the behavior of the system are known. Using dimensional analysis [1,2,3,4], an incomplete form of the characteristic equation of the system can be formulated. This equation is in terms of non-dimensional products of variables and parameters of the system. Then, similarity conditions can be established based on this equation.

Furthermore in 2005 an approach based on energy conservation was adopted to derive the law of scale for structural behavior [5]. The total potential energy of a scaled model and its full-scale counterpart must be proportional so that the conservation principle of the energy is satisfied. The applicability of the method is verified by deriving the law of scale both in the static and in the dynamic range in the case of beam and plate elements. The results showed that the derived scale factors can help to exactly predict the behavior of the prototype when the complete similarity requirements are fulfilled.

The applications of the laws of scale in question are numerous. In the aerospace field the scaling laws are applied for flutter prediction [6,7] and for study of vibrations [8,9].

In this research, these scaling laws were applied to the study of dynamic impact.

Few studies have focused on understanding the scaling behavior of transversely loaded plates, both under static and impact conditions [10,11]. Morton [12] investigated the use of dimensionless analysis applied to the (undamaged) elastic behavior of carbon/epoxy beams. He showed that the lay-up is important in assessing the likely validity of scale-model tests for such composites; rate effects are insignificant in the lay-ups and material system tested, and the only detectable rate effect occurred in one lay-up in which the flexural stiffness was matrix-dominated. Important size effects are noted as far as strength is concerned. Smaller specimens are always stronger than larger ones. This is thought to be due to the absolute size of matrix cracks and their effects on subsequent damage characteristics.

Nettles [13] used the same method to study quasi-static indentation and impact of unidirectional composite plates. Experimental results were compared to parameters obtained by similitude laws (contact force, displacement, damage area and indentation), and the differences were significant.

Quian et al. [14] introduced scale laws in the case of transversely impacted plates, using an analysis based on the equations of geometrically linear dynamic plates, and demonstrated that the impact response of plates subjected to linear deformations can be adequately scaled if damage in the specimen does not occur.

Ambur in his article [15] presented the applicability of the scaling laws to structural configuration with geometrically nonlinear deformations. In his paper, Ambur investigated the effect of scale on damage development.

Other authors have studied the scale and size effect on impact response.

Liu et al. [10] performed experimental tests on glass/epoxy laminates using an instrumented drop-weight impactor to estimate the impact response of composite laminates. Composite laminates of various in-plane dimensions and thicknesses were examined. The experimental results showed that in-plane dimensional effects are not as significant as the thickness effect, whereas the perforation is the most important damage stage in composite laminates subjected to impact loading since impact characteristics (peak force, contact duration and adsorbed energy) and mechanical property degradation (residual compressive maximum force and residual compressive absorbed energy) of composite laminates become stable one perforation takes place. However, the delamination played a very important role in the characterization of mechanical property degradation.

In 2008, P. Viot published an article [16] with the aim of introducing similitude techniques that can extend the responses of a real composite structure on a reduced scale. The experimental tests carried out concerned two sample scales by layering unidirectional carbon/epoxy layers. These tests showed that usual similitude laws can be used to predict the elastic behavior of a scaled structure.

The study of J. G. Carrillo [17] focuses on assessing the possibility of using scale-model tests for predicting the full-scale impact response of fiber-metal laminates (FMLs). The aim of his study was to investigate scaling effects in a novel fiber-metal laminate based on a polypropylene fiber reinforced polypropylene composite. Two approaches were used to investigate stacking sequence effects in scaling the FMLs, then the impact response of a FML was studied using a falling-weight impact tower. It was shown that the corresponding behavior hybrid follows a simple scaling law and that key parameters, such as the maximum impact force and the perforation energy, scale with size.

In 2011, M. Ramu in his article [18] developed similitude relations of elastic models. The establishment of similarity conditions, based on the Buckingham π theorem, is discussed and their use in the scaled model is also presented. The scale factor for the applied load and the corresponding interpretation of the results are also discussed [19,20].

The present research work is inspired by the experimental tests conducted published by P. Viot [16]. The idea is to develop a mathematical model capable of comparing samples of different comparison scales when subjected to dynamic impact.

The mathematical model reproduces the results of the similarity theory in the elastic field and extends to the analysis of destructive impacts, effectively opening a new phase of the investigation.

## 2. Scaling Problem Formulation

A similarity relationship between two systems is established by the Buckingham π theorem [4] which provides for a dimensional analysis of the characteristic parameters of a given system.

For a physical system it is possible to write Equation (1), which binds a parameter to all the remaining characteristics of the problem.
*f* (*X*_1_, *X*_2_, …, *X_k_*) = 0(1)

By applying the Buckingham π theorem, an equivalent equation can be rewritten, which describes the same physical phenomenon, but with a smaller number of dimensionless parameters, such as:
(2)$$g(\Pi_{1},\Pi_{2},\ldots ,\Pi_{m})=0$$

The number of physical variables *k* and that relating to independent dimensions *q* vary from case to case and must be estimated at each application of the theorem. As a rule, the method of using it is quite simple and mechanical. The main steps are highlighted below.

Create a list of all the independent physical variables that govern the system, then estimate the number *k*.Find the number of independent dimensions and estimate the number *q*. In most applications in the structural field, the independent dimensions are mass [*M*], length [*L*] and time [*T*].Assign independent dimensions to the physical variables identified in step 1.Select from the physical variables a number *q* of parameters which, together, include all independent dimensions.Calculate the m dimensionless parameters, imposing that each of the variables taken from the group of those not selected at point 4 (i.e., the unselected *k* ≠ *q*), multiplied by the *q* selected variables, each raised to a natural number unknown *α_j_* with *j* = 1, 2, …, *r*, results in a dimensionless number.

For each of the *m* unselected variables, a system of *q* equations in the variables α*_j_* is obtained, which characterizes the m dimensionless parameters Π*_i_* [21]. Equation (2) can also be rewritten by making explicit one of the dimensionless parameters, for example the first.
(3)$$\Pi_{1}=\emptyset (\Pi_{2},\Pi_{3},\ldots ,\Pi_{m})=0$$

If the complete similarity is valid, that is, the set of all similarities, the dimensionless value of Π_1*r*_ relative to the real system must coincide with the same dimensionless parameter relative to the scaled system Π_1*s*_. Therefore:(4)$$\frac{\Pi_{1s}}{\Pi_{1r}}=\frac{\emptyset \left(\Pi_{2s},\Pi_{3s},\ldots \Pi_{ms}\right)}{\emptyset \left(\Pi_{2r},\Pi_{3r},\ldots \Pi_{mr}\right)}=1$$

Given the equality of the function ∅,  it can be deduced that:(5)$$\{\begin{array}{c} \Pi_{1s}=\Pi_{1r}\\ \Pi_{2s}=\Pi_{2r}\\ .\\ .\\ .\\ \Pi_{ms}=\Pi_{mr} \end{array}$$

From the Equation (5) assigned the scale factors of the *q* physical parameters selected in point 4, all the scales relating to the remaining parameters are obtained.

The applicability of this procedure is closely linked to the equality between the variables of the system to be studied and those of the scale model. However, there is also another way to develop a law of similarity or directly use the equations that govern the system. Thus, it is not necessary to have all the parameters characterizing the system (which in the previous case are transformed into dimensionless parameters), but it is essential to have a good knowledge of the equations and relationships underlying the two phenomena to be compared. Therefore, a set and the scaled model are two different systems responding to a similarity, but not necessarily with identical parameters. By virtue of this, the nature of any set is to be modeled mathematically, obviously as a function of its variables and parameters.

Given the mathematical relationships that express the characteristics and functioning of a full-scale prototype, the necessary and sufficient conditions for a similarity relationship with a scaled model to be valid require that the mathematical model of the scaled system be transformed into that of the prototype (or vice versa) through a one-to-one relationship. In other words, starting from the knowledge of the way in which a given real system replies to a specific input, therefore knowing the output of the system, one can predict with adequate accuracy the output of all systems like it, provided that an input like that applied to the real system is applied to these. This means that, with {*X_r_*} and {*X_s_*} being the vectors of the characteristic outputs of the real system and of the scaled model, the relationship between these two is regulated by the diagonal transformation matrix [Λ]*_sr_*, such that:(6)$$\left\{X_{r}\right\}={\left[\Lambda \right]}_{sr}\left\{X_{s}\right\}$$
(7)$$\left\{X_{s}\right\}=\left[\Lambda_{sr}^{-1}\right]\left\{X_{r}\right\}$$
where the elements of the generic vector *X* represent the variables and parameters of the system. The diagonal elements of the matrix [Λ]*_sr_* are none other than the scaling factors of the relevant elements of the characteristic vector *X*:(8)$$\left[\begin{array}{cccc} \lambda_{x1} & 0 & \cdots & 0\\ 0 & \lambda_{x2} & \cdots & 0\\ \vdots & \vdots & \ddots & \vdots \\ 0 & 0 & \cdots & \lambda_{xn} \end{array}\right]$$

The generic *i*-th scale factor is also determined.
(9)$$\lambda_{Xi}=\frac{X_{ri}}{X_{si}}$$

To establish the similarity conditions between the real system and the scaled prototype, two procedures can be used, the application of the Buckingham π theorem and the direct use of system equations. Similarity conditions can be obtained directly from the system’s field equations or via Buckingham theorem (which is a dimensional analysis of all effects) when the mathematical model is not available. In this case all variables as well as system parameters must be known. Using the dimensional analysis approach, an incomplete form of the characteristic equation of the system may be sufficient, but it is necessary to know in advance all the parameters that influence the physical phenomenon [21]. The approach of dimensioning system equations, on the other hand, could be more convenient than dimensional analysis since the resulting similarity conditions are more specific. When the equations governing the system are used to establish the conditions of similarity, the relationships between the variables are directly established by the equations themselves. The field equations of a system, equipped with the correct boundary conditions and initial conditions, characterize the behavior of the system in terms of variables and parameters. If the field equations of the scale model and the real system are invariant with respect to the transformation $\left[\Lambda_{sr}\right]$ and ${\left[\Lambda \right]}_{sr}^{-1}$, then between the two systems there is a complete similarity. This transformation characterizes for two systems the similarity conditions between all the structural geometry parameters and the response of the two systems. It is convenient to present an example capable of fully clarifying the concept behind the dimensioning of the system equations. Consider a laminated plate of the “beamplate” type, consisting of orthotropic material, of width “*a*” in the “*x*” direction, of infinite length in the “*y*” direction and simply resting at the two opposite ends, as in Figure 1.

To find the maximum deformation of the plate when it is subjected to a transversal load, that is a load acting on a line parallel to the “*y*” axis and in a central position with respect to the width, having the dimensions of a force divided by a length. Assuming that the displacement functions are independent of “*y*”, or equivalently if *u* = *u* (*x*), *v* = 0, *w* = *w* (*x*), the plate assumes a cylindrical bending, and bearing in mind the solution of Ashton and Whitney [22], the differential equations governing bending are reduced to:(10)$$\{\begin{array}{c} \frac{d^{4}\omega }{dx^{4}}=\frac{qA_{11}}{\left(A_{11}D_{11}-B_{11}^{2}\right)}\\ \frac{d^{3}u}{dx^{3}}=\frac{B_{11}}{\left(A_{11}\right)}\frac{d^{4}\omega }{dx^{4}}\\ \omega =0\\ N_{xx}=A_{11}\frac{du}{dx}-B_{11}\frac{d^{2}\omega }{dx^{2}}=0\\ M_{xx}=B_{11}\frac{du}{dx}-D_{11}\frac{d^{2}\omega }{dx^{2}}=0 \end{array}$$
where the terms *A*_11_, *B*_11_ and *D*_11_ are the terms contained in the matrix that binds the loads with the deformations of a composite laminate. Considered a real system, Equation (10) can be rewritten as:(11)$$\left(A_{11r}D_{11r}-B_{11r}^{2}\right)\frac{d^{4}\omega_{r}}{{dx}_{r}^{4}}={qA}_{11r}$$
(12)$$\left(A_{11r}D_{11s}-B_{11s}^{2}\right)\frac{d^{4}\omega_{r}}{dx_{s}^{4}}=qA_{11s}$$

The similitude relationship is written that links the parameters relative to the real system with those of the scale model.
(13)$$\left\{\begin{array}{c} A_{11r}\\ B_{11r}\\ D_{11r}\\ q_{r}\\ x_{r}\\ \omega_{r} \end{array}\right\}=\left[\begin{array}{cccccc} \lambda_{A_{11}} & 0 & 0 & 0 & 0 & 0\\ 0 & \lambda_{B_{11}} & 0 & 0 & 0 & 0\\ 0 & 0 & \lambda_{D_{11}} & 0 & 0 & 0\\ 0 & 0 & 0 & \lambda_{\omega } & 0 & 0\\ 0 & 0 & 0 & 0 & \lambda_{x} & 0\\ 0 & 0 & 0 & 0 & 0 & \lambda_{q} \end{array}\right]\left\{\begin{array}{c} A_{11s}\\ B_{11s}\\ D_{11s}\\ q_{s}\\ x_{s}\\ \omega_{s} \end{array}\right\}$$

Substituting the values of Equation (13) into Equation (11) leads to the following:(14)$$\frac{\lambda_{A_{11}}\lambda_{D_{11}}\lambda_{\omega }}{\lambda_{x}^{4}}A_{11s}D_{11s}\frac{d^{4}\omega_{s}}{dx_{s}^{4}}-\frac{\lambda_{B11}^{2}\lambda_{\omega }}{\lambda_{x}^{4}}B_{11s}^{2}\frac{d^{4}\omega_{s}}{dx_{s}^{4}}=\lambda_{q}\lambda_{A11}qA_{11s}$$

If the similarity between the two systems is valid then Equation (14), which governs the real system, must be identical to that relating to the scaled model Equation (12). Therefore, all dimensionless groups have to be equal to unity.
(15)$$\frac{\lambda_{A_{11}}\lambda_{D_{11}}\lambda_{\omega }}{\lambda_{x}^{4}}=\frac{\lambda_{B11}^{2}\lambda_{\omega }}{\lambda_{x}^{4}}=\lambda_{q}\lambda_{A11}=1$$

By applying the same reasoning to the other equations of the system, all the relations necessary to complete the conditions of similarities between the full-scale prototype and the reduced model are obtained.

With the expression “*scaling effects*”, it is usual to define the effects that occur on the response in the field of elastic deformation due to external causes (any type of force: static, dynamic, concentrated or distributed) when a change of scale is performed. With the term “*size effects*”, on the other hand, there are the effects that the change in size of a structure causes to the formation and development of damage, and in presence of the composite materials the technological process is to be considered along with the environment and the production temperature of the studied structure [23]. In many scientific works, in researching the conditions of similarities under which to obtain similarities in the elastic response, the terms “scaling effects” and “size effects” have the same meaning, since the interest is the evaluation of the damage to the structure. The most relevant studies on the effects that scaling produces on the stiffness in the elastic response have been carried out by Bazant and Rajapakse [24] who conclude that the effects of scaling on the stiffness of a structure appear to be fundamentally negligible. As for the effect of the change in geometric scale on the maximum load that can be supported (strength) by a structure, more specific studies are still needed, since the value of the aid of the similarity theory, in case of failure, is not yet well known [25]. However, G. Grimes [23] concluded that for laminated solids the larger scale effect on mechanical resistance properties in the static case is less than 4.5% and that the effects of scale are not due to size but to the procedures and quality of the production of two similar structures. The study of scale effects is therefore a very complex operation and represents the focal point of the limits of the theory of similarity.

## 3. Experimental Impact Tests

To validate the use of the similarity theory in the field of elastic deformation on the one hand and to evaluate the effects of scale on the structures that reach damage on the other, the experimental tests from Viot et al. [16] were investigated, as reported in the Figure 2.

The declared objective was to demonstrate the validity and applicability of the similarity laws for two flat carbon fiber plates, subjected to bending during impact with two hemispherical shape impactors geometrically scaled with the same scale factor used for the plates. As shown in the last figure, in case “*A*” the panel and the dart were geometrically scaled by a factor of 1:2 compared to case “*B*”. This means that in case “*B*”, the panel and the dart had doubled characteristic lengths compared to those of case “*A*”. The behavior of these panels was therefore analyzed, and the results of the theory of similarity, obtained by applying the Buckingham π theorem, were used to deduce the conditions and limits of applicability of the theory.

Morton [12] analyzed the effects of scale, using the dimensional analysis applied to the case of beams subjected to a dynamic load caused by an impact. In the case of isotropic beams not subject to damage, it is possible to identify 13 characteristic parameters of the phenomenon in question:I.***l***, the length of the rafters.II.***h***, the thickness of the beams.III.***b***, the width of the beams.IV.***w***, the deflection at the center of the beams.V.***R_i_***, the radius of the hemispherical dart.VI.***E***, Young’s modulus of the material constituting the beams.VII.***ν***, Poisson’s coefficient of the material of the beams.VIII.***ρ***, the density of the beams.IX.***E_i_***, Young’s modulus of the dart material.X.***ν_i_***, Poisson’s coefficient of the dart material.XI.***ρ_i_***, the density of the dart.XII.***V_i_***, the speed of the darts at the impact with the beams.XIII.***t***, time.

Although this study formally deals with a phenomenon different from that concerning composite plates due to the different nature of the materials, it can also be applied to the case of plates without affecting their generalities. Thus, the number of parameters to be used is lower than in the case of plates.

Considering that *k* = 13 (the independent physical variables) and *q* = 3 (the number of independent dimensions), identified as mass [*M*], length [*L*] and time [*T*], then the independent dimensions are assigned to each of the *k* characteristic parameters are:


[*l*] = [*L*]
[*h*] = [*L*]
[*b*] = [*L*]
[*w*] = [*L*]
[*R_i_*] = [*L*]
[*E*] = [*M*][*L*^−1^][*T*^−2^]
[*ν*] = [*M*^0^][*L*^0^][*T*^0^]
[*ρ*] = [*M*][*L*^−3^]
[*E_i_*] = [*M*][*L*^−1^][*T*^−2^]
[*ν_i_*] = [*M*^0^][*L*^0^][*T*^−2^]
[*ρ_i_*] = [*M*][*L*^−3^]
[*V_i_*] = [*L*][*T*^−1^]
[*t*] = [*T*]

The dimensions *h*, ρ and *E* include all independent dimensions Finally, *m* = *k* − *q* = 10 non-dimensional characteristic parameters were calculated, as follows:(16)$$\Pi_{1}=\omega \;\left(h^{\alpha 1}\rho^{\alpha 2}E^{\alpha 3}\right)$$
$$\left[\Pi_{1}\right]=\left[L\right]\;\left(\left[L^{\alpha 1}\right]\left[M^{\alpha 2}\right]\left[M^{-3\alpha 2}\right]\left[M^{\alpha 3}\right]\left[L^{-\alpha 3}\right]\left[T^{-2\alpha 3}\right]\right)=\left[M^{0}\right]\left[L^{0}\right]\left[T^{0}\right]$$

By equating the exponents, the following is obtained:



$$\alpha_{1}=-1$$





$$\alpha_{2}=0$$





$$\alpha_{3}=0$$



Therefore, by replacing these latter relations in Equation (16) the following is obtained:



$$\Pi_{1}=\frac{\omega }{h}$$



Using the same procedure for all (*k* − *q*) parameters, Table 1 is completed:

It is necessary to assign the value of the scale factors to be used for the three selected physical parameters (*h*, ρ and *E*) so that the remaining factors can be calculated. Therefore, if the geometric scale ($\lambda_{h}=$ λ) is fixed and the correspondence between the materials of the beams and the bolts is verified, such that $\lambda_{p}=\lambda_{E}=\lambda^{0}$, it is enough to replace these ratios in Equation (6) and make use of Table 1 to extract all the remaining scale factors relating to the m characteristic parameters of the system. They are presented in the Table 2, considering that all the scale factors relating to the other characteristic quantities can be easily obtained with the aid of dimensional analysis.

The scale factor of the area can be obtained as the product of the scale factors of the dimensions along the two axes that form the plane in which the affected area lies. In our case an undistorted geometric simile was used: $\lambda_{A}=\lambda^{2}$;The volume scale factor can be obtained as the product of the scale factors of the dimensions along the three main axes of the plate: $\lambda_{\nu }=\lambda^{3}$;The scale factors of the mass $\lambda_{m}$ are obtained by multiplying the scale factor of the volume by the scale factor of the density, obtaining: $\lambda_{m}=\lambda_{\nu }\lambda_{\rho }$.The scale factor relative to the velocity has been previously calculated as: $\lambda_{\nu }=\lambda^{0}$;The scale factor of the acceleration is obtained starting from that of the speed divided by that relating to time: $\lambda_{a}=\frac{\lambda_{V}}{\lambda_{t}}$;The scale factor of the force $\lambda_{f}$ is obtained by multiplying that of the acceleration by that of the mass: $\lambda_{f}=\lambda_{m}\lambda_{ai}$;For the energy scale factor $\lambda_{U}$, with obvious meaning of the symbols, the following is obtained: $\lambda_{U}=\lambda_{f}\lambda$;Finally, the previously calculated time scale factor is $\lambda_{t}=\lambda$.

Table 3 summarizes, therefore, the scale factors of the characteristic quantities of the systems constituted by the beams or equivalently by the plates and by the bolts.

The experimental tests were carried out with a drop tower [16]. As far as the specimens are concerned, two flat plates were manufactured differing only in geometric scale. The samples were made by unidirectional carbon/epoxy pre-preg 914 34% foils, i.e., pre-impregnated foils in carbon fibers held together by an epoxy resin matrix in a volume fraction equal to 34%. The thickness of the single sheet is approximately 125 μm, while the final thickness is 1.5 mm with a lamination sequence of the first specimen (sample “*A*”) equal to ${\left[0_{2}^{{}^{\circ}}/90_{3}^{{}^{\circ}}/0^{{}^{\circ}}\right]}_{s}$. The geometric characteristics of the second specimen (sample “*B*”) were determined using the similarity laws and with a geometric scale factor λ=2. Therefore, the thickness of laminate B is 3 mm, obtained by doubling the thickness of each single sheet through the “sub-ply level scaling” method reported in Reference [25], obtaining the following lamination sequence. ${\left[0_{4}^{{}^{\circ}}/90_{6}^{{}^{\circ}}/0_{2}^{{}^{\circ}}\right]}_{s}$.

The characteristics of the manufacturing process related to sample “*A*” and “*B*” are identical.

The impact tests are performed considering two different speeds for each of the two samples, 1.8 and 2.2 m/s, with the aim of estimating the validity of the similarity laws and describing the behavior of the composite material. The lowest velocity can avoid visible damage to the naked eye. The velocity of 2.2 m/s can achieve significant damage on at least one of the two scales. It was thus possible to first measure the response of the structure according to damage speed and secondly to estimate the relevance of the similarity laws in both cases (with or without the presence of damage). The composite plates were placed on two cylindrical supports, whose axes are directed according to the direction of side a of the plates and coplanar with the plane of the plates, with a diameter of about 4 cm, to obtain a dynamic test of “three-point bending”. The distance between the supports was set according to the dimensions of the plate and the conditions imposed by the laws of similarity. The distance d between the supports is 100 mm for sample “*A*” and consequently 200 mm for sample “*B*”. The other test parameters were determined starting from the scale factors found (see Table 4); the diameters of the hemispherical darts for sample “*A*” and “*B*” were scaled by a factor λ, and the hemispherical impacting body of sample “*B*” was therefore twice as large as that of sample “*A*”. Consequently, since the density between the two cases must be in unit ratio, the mass of the impacting body is proportional to $\lambda^{3}$, and it is therefore 8.6 kg for sample “*B*”, i.e., eight times the mass of the dart in sample “*A*” (1.075 kg).

This approach was chosen to compare both the trend of the dart’s displacement as a function of time and that relating to the contact force between plate and dart as a function of the displacement. The next graphs are related to the experimental results obtained for sample “*B*” with those obtained by scaling sample “*A*” according to the scale factor $\lambda^{i}$ (with *i* = 1, 2). The first comparison concerns the tests performed at an impact speed of 1.8 m/s; during the tests the specimens exhibit no damage, and the behavior can be considered with an excellent elastic approximation. The curve λA deduced from the theory of similarity is quite adherent to that obtained from the tests performed on panel “*B*”. In particular, the points of maximum deflection are completely superimposable, and any possible error is practically negligible. As for the contact time of case “*A*”, multiplied by the corresponding scale factor λ, it is noted how it is slightly lower than that relating to case “*B*”: the calculated error is about 2%.

Both images reported in Figure 3 show the similarity conditions in the field of the elastic response, and it is equally evident that when the dart impacts with a speed of 2.2 m/s, visible damage is generated on the tested panel.

For the case of sample “*B*”, the graph shows the displacement of the dart as a function of time, comparing the curves drawn experimentally with those obtained by scaling panel “*A*” by a scale factor λ. In the first phase of the impact (section I–M), the two curves are very close because the material has not yet broken; in proximity of the elastic field there is an appreciable correspondence with the laws of similarity. It should be noted, however, that point M of maximum deflection is slightly higher than the one actually obtained, M’; the error (about 3%) is like that obtained between sample “*A*” and sample “*B*” for the lowest impact speed. Once delamination has started, sample “*B*” changes its characteristics and loses stiffness, presenting local defects and “transforming” into a new structure. This is due to an evident distancing between the two curves: During the ascent of the impacting dart (section M’–K’), curve λ A does not follow the actual changes that sample “*B*” undergoes because of the break. By virtue of this, even the calculated contact time (24 ms) is much less than the real 37 ms. Observing the slope of the curves in the ascent section (K’–L’), it is clear how the rigidity of the structure is influenced by the phenomenon of the destructive impact: In the real sample an evident loss of flexion stiffness is visible, which involves a lower slope of the lift curve. This confirms what has been stated up to now, namely that in the field of destructive impact the laws of similarity do not give reliability in the description of the phenomenon.

The abrupt breaking of the material defines the progressive delamination of the sheets, which drastically reduces the flexural stiffness of plate “*B*”, generating an immediate reduction of the load acting on the panel and a consequent increase in the contact time.

As a first observation from the Figure 4, the graph for sample “*A*” shows a strong oscillation of the contact force during the impact for both speeds: The appreciable vibrations are due to the proper modes of vibration of the plate, generated at the moment in which it occurs the impact (point “J” on the displacement graph). Near point M of maximum deformation, the oscillations show a strong attenuation.

Lower impact speed, for both samples (“*A*” and “*B*”), is very small when compared with the kinetic energy of the initial dart. This is indicative of the fact that the structural damage caused is extremely small. As regards the second experimental test carried out on sample “*B*”, the impact at the lowest speed, there are no substantial differences with the test on sample “*A*”, since the increase in the contact force proportional to the trend of the impact energy. The difference lies in reaching the maximum value of the peak force, which as expected is higher for sample “*B*” (2265 N).

The high energy impact behavior is interesting; for the high-speed test, the contact force increases linearly as a function of the displacement until it reaches the maximum peak of 2730 N. Once this value of the maximum force is reached, delamination begins. The damage is increasingly extensive, and this causes a sudden drop in the contact force. Despite the strong vibrations to which the panel is subjected, the structure can support up to a load of 1000 N.

## 4. Numerical Approach for Low Velocity Impact

The experimental tests conducted by P. Viot et al. verified a discrepancy between the similarity theory and the experimental results obtained. The objective of this study is very similar to that at which the experimental tests intended to arrive, verifying how close the results of the numerical simulations are to the similarity theory and obtaining a coefficient to be applied to the kinetic energy of impact of a scaled structure to make the two systems equivalent from the point of view of the break.

Initially, emphasis was placed on the search for the best setup so that the results of the simulations were consistent with the experimental tests and, on the other hand, to compare the same results with the similarity theory to evaluate a possible effect of the failure to scale the constraints. This calibration phase of the simulations is also necessary to obtain valid information on the behavior of the material models and formulations. Therefore, it was decided to carry out a set of simulations that differ in three different characteristics, which are significant for the purposes of the simulations.

The type of constraint was considered along with the most appropriate orthotropic material model to also implement damage to the material (MAT 022, MAT 054 and MAT 058) and finally the formulation of the SHELL element: the classic Belytschko–Tsay formulation (2), the Belytschko Leviathan formulation (8), the Belytschko–Wong–Chiang formulation (10) and the fully integrated (16).

To reach the objective that has been set, it is enough to consider only the impact tests that did not induce considerable damage; therefore, the results were compared with the case of impact velocity equal at 1.8 m/s. Therefore, the total number of tests to obtain a correct calibration of the numerical setup is 2 × 3 × 4 = 24 for case “*A*” and another 24 for case “*B*”.

Only subsequently, once the best method to simulate this type of impact has been ascertained as reported in the Figure 5, is a confirmation obtained with the experimental tests regarding the damage to plates “*A*” and “*B*”; subsequently there are variations in the speed of impact and therefore in the kinetic energy of the dart (or equivalently of the deformation energy absorbed by the plate), with the aim of obtaining a coefficient to be applied to the scaling factor of energy.

The Table 5 and Table 6 show the parametric analysis corresponding to the different conditions, where:*W_max_* reports the values of maximum displacement in the middle;*F_cmax_* is the maximum contact force;*F_cfmax_* is the maximum contact force filtered with an average of five points;*t_c_* is the contact time measured as the time between the first and the last non-zero contact force value;*U* is the residual internal energy, calculated as the minimum internal energy downstream of the maximum point;*EH* and *EH_u_* consider the maximum hourglass energy and the uncontrolled hourglass energy of the simulations that exhibited a maximum hourglass energy value over 10% of the kinetic energy initially possessed by the dart, which is not considered.

### 4.1. Comparison of the Results with Experimental Tests

In this regard, considering that no type of numerical damping has been added, and given the hypothesis of quasi-linear behavior of the material up to failure, it is expected that the deformation energy (or internal energy) accumulated during the loading phase inside the plate, it is completely returned in the unloading phase, obviously if the structure does not damage. In addition, it presents an amount of energy associated with the hourglass effect representing a numerical defect, for which a positive judgment can be attributed when this energy remains low when compared with the total energy of the system, and vice versa. The symbol *T* denotes the results of the experimental tests and the symbol *k* the values of the quantities obtained through the averaging operation carried out on groups of simulations having the *k*-th characteristic in common. The symbol *k* represents the average of a specific type of constraint, the average of all simulations relating to a specific material model or element formulation.

The following ratios indicate the percentage variations of the averages for the *k*-th characteristics of the maximum displacement *W^k^_max_* of the maximum contact force *F^k^_cmax_* and of the contact time *At^k^_c_* with respect to the same values obtained from the experimental tests.
(17)$$V_{W}^{kT}=\frac{W_{max}^{k}}{W_{max}^{T}}-1$$
(18)$$V_{F}^{kT}=\frac{F_{cmax}^{k}}{F_{cmax}^{T}}-1$$
(19)$$V_{\Delta t_{c}^{k}}^{kT}=\frac{\Delta t_{c}^{k}}{\Delta t_{c}^{T}}$$

The next ratios indicate, on the other hand, the percentages of accumulated internal energy $\Delta U^{k}$ and maximum energy of hourglass $\Delta_{max}^{k}$ x with respect to the total energy of the *E_T_* system.
(20)$$V_{\Delta U^{k}}^{kT}=\frac{\Delta U^{k}}{E_{T}}$$
(21)$$V_{H_{max}^{k}}^{kT}=\frac{H_{max}^{k}}{E_{T}}$$

Furthermore, to carry out an evaluation of the results, an *ID^kT^* distortion index is proposed, which highlights the difference of a numerical result obtained with a given method, compared to the experimental result. For this reason, it was decided to set the *ID^kT^* index equal to:(22)$$ID^{kT}=\frac{1}{5}\left(\left|V_{W}^{kT}\right|+\left|V_{F}^{kT}\right|+\left|V_{\Delta t_{c}}^{kT}\right|+\left|V_{\Delta U}^{kT}\right|+\left|V_{Hmax}^{kT}\right|\right)$$

Therefore, according to the distortion index taken into consideration in the Table 7, the configuration that most adheres to the experimental tests is that relating to the use of cylindrical supports, the material model MAT 054 and the Belytschko–Wong–Chiang formulation as underlined in the Table 7.

### 4.2. Verification of the Similarity Theory

As with for the experimental tests, it is possible to evaluate the percentage variation of the ratios between the parameters *W_max_*, *F_cmax_*, *F_cfmax_*, *t_c_* and *U* relative to the case of full-scale plate “*B*” and those relating to the case of reduced plate *A*, with respect to the scale factors of the specific quantity, simply by placing
(23)$$V_{W}^{\left(BA\right)}=\frac{W_{maxB}}{\lambda W_{maxA}}-1$$
(24)$$V_{F}^{\left(BA\right)}=\frac{F_{cmaxB}}{\lambda^{2}F_{c\;maxA}}-1$$
(25)$$V_{Ff}^{\left(BA\right)}=\frac{F_{cf\;maxB}}{\lambda^{2}F_{cf\;maxB}}-1$$
(26)$$V_{\Delta t_{c}}^{\left(BA\right)}=\frac{\Delta t_{cB}}{\lambda \Delta t_{cA}}-1$$
(27)$$V_{\Delta U}^{\left(BA\right)}=\frac{\Delta U_{B}}{\lambda^{3}\Delta U_{A}}-1$$

To evaluate which configuration most respects the similarity theory; as detailed in the Table 8, the same procedure is considered, introducing the index as:(28)$$ID^{kBA}=\frac{1}{4}\left(\left|V_{W}^{kBA}\right|+\left|V_{F}^{kBA}\right|+\left|V_{Ff}^{kBA}\right|+\left|V_{\Delta t_{c}}^{kBA}\right|\right)$$

According to the *ID^kBA^* distortion index, unlike the *ID^kT^* index, it provides a greater adherence to the similarity theory using the setup consisting of the use of numerical supports (SFSF), of the material MAT 058 and a fully integrated formulation for SHELL elements, as reported in Figure 6.

In the previous paragraph two points were substantially clarified:

The constraint condition adopted by the experimental tests has an influence on the scaling theory.The simulations carried out with the chosen setup also showed excellent adherence with the similarity theory as well as being consistent with the experimental results.

### 4.3. Energy of Destructive Impact

Using this type of numerical configuration, the results of the simulations carried out with a higher impact speed equal to 2.2 m/s can be used to investigate the experimental evidence showing delamination damage. The damage contours of the simulations of case “*A*” and “*B*” are shown in Figure 7.

With the material model MAT 054 it is possible to visualize the internal damage to each element; the solver associates the value 1 with the integration points that satisfy the failure criterion and associates the value 0 with the other integration points that still work in the field linear elastic. Through an average operation on the integration points, the solver associates a value with each element, indicative of the damage created within the element. Note that plate “*B*” lost its curvature on the surfaces to the right and left of the dart and therefore lost flexural stiffness, which suggests that the plate has suffered a break, behaving, in fact, like a membrane. Figure 8a shows the curves of the displacement of the central node, Figure 8b the curves of the contact force at the interface, Figure 8c those contact forces as a function of the displacement of the central node and finally Figure 8d the internal energy and hourglass graphs. Note that the breaking of the plate first causes a peak in the contact force, then the plate and the dart lose their contact and the force settles at zero.

### 4.4. Search for Equivalent Damage

The last step proposes a method to predict the failure of a panel scaled by a certain scale factor λ starting from the knowledge of the failure behavior of a second panel. The method used envisages the aid of a multiplicative coefficient Ω such that multiplied by the scale factor, it returns a correct one to be used in the phenomena of destructive impacts on flat composite panels to correctly scale the initial kinetic energy possessed by the impacting body. That is, indicating with *E_cA_* and *E_cB_*, respectively, the minimum kinetic energy of the dart until plates “*A*” and “*B*” exhibit a loss of flexural stiffness, the following arises:(29)$$\frac{E_{cB}}{E_{cA}\lambda^{3}\Omega^{3}}=1$$

From this equation the formula relating to the corrective coefficient is:(30)$$\Omega ={\left(\;\frac{E_{cB}}{E_{cA}\lambda^{3}\Omega^{3}}\right)}^{\frac{1}{3}}$$

The relationship between the minimum destructive energy turns out to be:(31)$$\frac{E_{cB}}{E_{cA}}=\frac{V_{iB}^{2}}{V_{iA}^{2}}\frac{m_{iB}}{m_{iA}}={\left(\frac{V_{iB}^{2}}{V_{iA}^{2}}\right)}^{2}\lambda^{3}$$
where for simplicity *V_iA_* and *V_iB_* indicate the impact velocities corresponding to the respective minimum destructive kinetic energy and *m_iA_* and *m_iB_* the masses (also scaled) of the impacting darts. Finally, replacing Equations (30) and (31), the following is obtained:(32)$$\Omega ={\left(\frac{V_{iB}^{2}}{V_{iA}^{2}}\right)}^{\frac{2}{3}}$$

Accordingly, to find this value, a set of 16 simulations were considered, eight for each case. Both in case “*A*” and in case “*B*”, the impact simulations at the speed *V_i_* = 1.8 m/s were considered, increasing the speed by 0.1 m/s in each pair of subsequent simulations, up to the speed of 2.5 m/s, determining the presence of a structural failure. Table 9 summarizes the results obtained from the 16 simulations with the terms: failure, i.e., “*failure*” of the panel, and “*elastic*”, which represents an almost elastic behavior of the structure.

The minimum impact speed for there to be a failure was found to be equal to 2.4 m/s for plate “*A*” and 2.2 m/s for plate “*B*”. Therefore, using Equation (32) the value calculated in the case of a scale factor equal to λ = 2 is:(33)Ω=0.944

It is immediately clear that this coefficient is not constant with the variation of the scale value because the ratio between the impact velocities of the two cases, such as to cause the panels to collapse, also varies with scale factor. For a unit scale factor (equal panels), the ratio between the speeds is also unitary and can only be unitary.

## 5. Conclusions

This paper reports the results of experimental tests present in the bibliography [16] on samples of different scales subject to low velocity impact at different energy levels. From these experimental data, a numerical model was created, which aims to extend the analysis of scale factors to structural applications of nonlinear dynamics that lead to the failure of the structure in question.

The results of the similarity theory were consistent with both the experimental tests and the numerical simulations performed by means of the commercial solver LS-Dyna in the elastic field. For impact energy values lower than the critical ones, the results of the simulations were confirmed to be always in line with the experimental tests, showing deviations of less than 10% for each characteristic parameter considered.

In presence of the destructive impact, an adherence with the experimental tests was maintained. This work led to the definition of a corrective coefficient (λ) formulated for the geometric scale factor to be used for the calculation of the impact velocities that produce structural failure conditions of flat panels in composite material. At the same time, it was demonstrated that for scale factors >2, the ratio between the deformation speeds assumes such values that the effects on the materials cannot be neglected.

## Figures and Tables

**Figure 1 materials-14-05884-f001:**
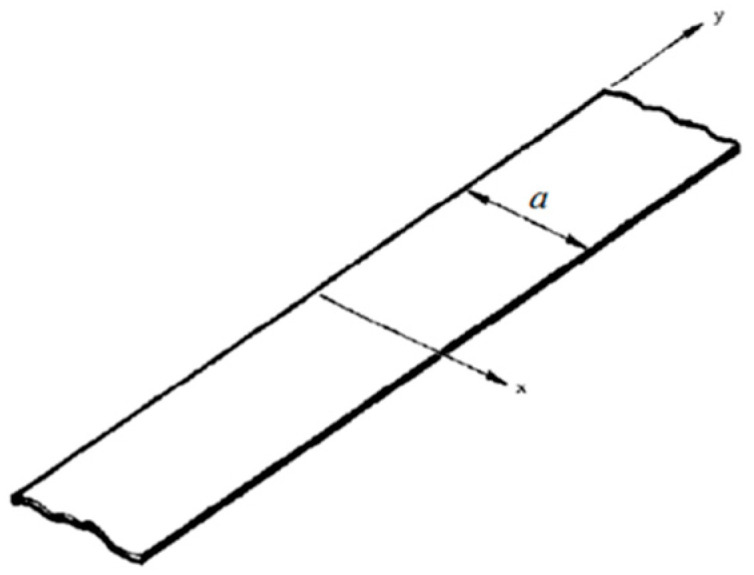
Logical scheme of similarities.

**Figure 2 materials-14-05884-f002:**
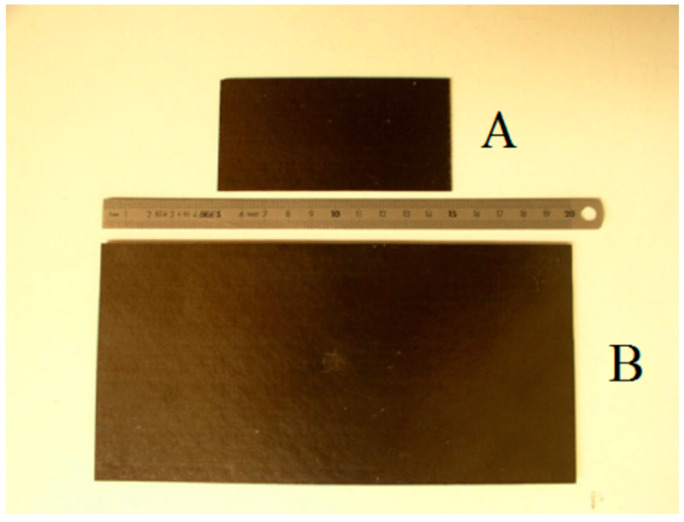
Samples in the experimental tests [16].

**Figure 3 materials-14-05884-f003:**
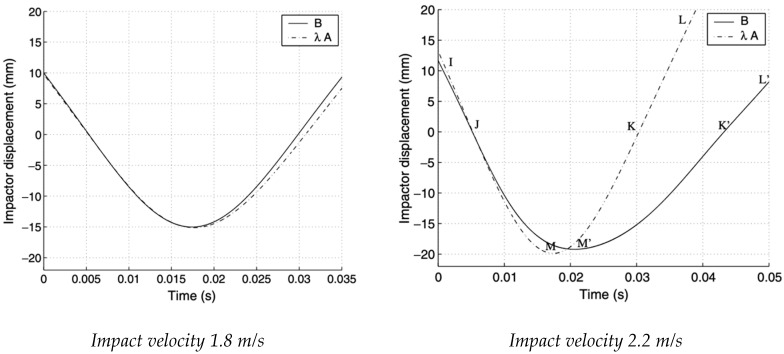
Displacement of the hemispherical dart, at speeds of 1.8 and 2.2 m/s [16].

**Figure 4 materials-14-05884-f004:**
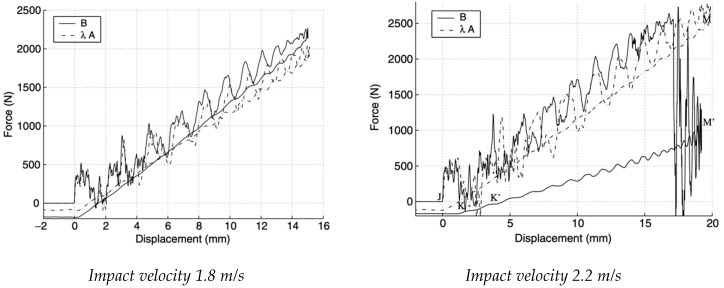
Force–displacement, at speeds of 1.8 and 2.2 m/s [16].

**Figure 5 materials-14-05884-f005:**
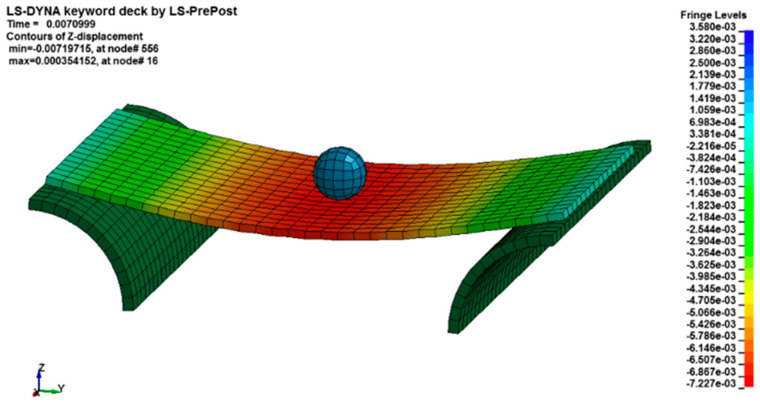
Maximum displacement for the plate during the impact evolution.

**Figure 6 materials-14-05884-f006:**
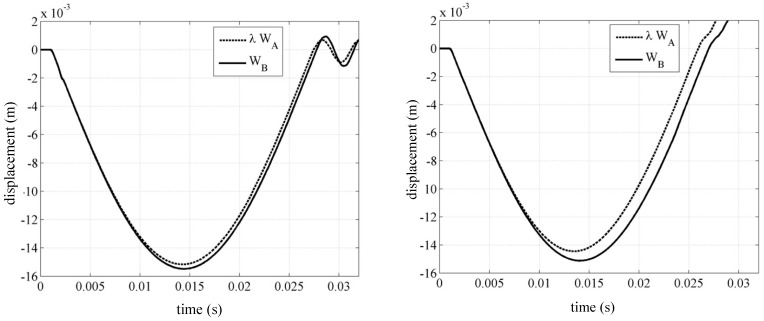
Time-history of the displacement of the central node as a function of time.

**Figure 7 materials-14-05884-f007:**
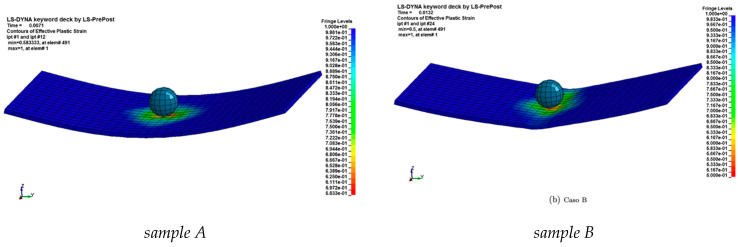
Maximum displacement for the plate during the impact evolution.

**Figure 8 materials-14-05884-f008:**
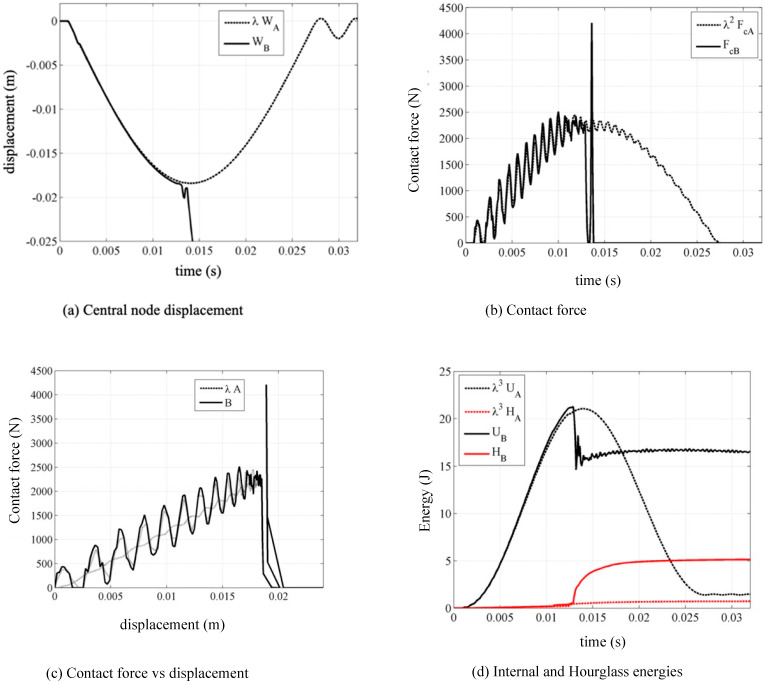
Comparison of the curves of case B with the scaled case “A”.

**Table 1 materials-14-05884-t001:** Dimensionless parameters.

**Geometric Parameters**	$\Pi_{1}=\frac{\omega }{h}$	$\Pi_{2}=\frac{l}{h}$	$\Pi_{3}=\frac{b}{h}$	$\Pi_{4}=\frac{R_{i}}{h}$
**Materials Parameters**	$\Pi_{5}=\frac{E_{i}}{E}$	$\Pi_{6}=\nu$	$\Pi_{7}=\nu_{i}$	$\Pi_{8}=\frac{\rho_{i}}{\rho }$
**Test Conditions**	$\Pi_{9}=\frac{\rho_{i}V_{i}^{2}}{E}$	$\Pi_{10}=\frac{tV_{i}}{h}$		

**Table 2 materials-14-05884-t002:** System parameter scale factor table.

Parameters	Scale Factor
*l*	λ
*b*	λ
ν	λ
$R_{i}$	λ
$\nu_{i}$	$\lambda^{0}$
$E_{i}$	$\lambda^{0}$
$v_{i}$	$\lambda^{0}$
$\rho_{i}$	$\lambda^{0}$
$V_{i}$	$\lambda^{0}$
*t*	λ

**Table 3 materials-14-05884-t003:** Scaling factors of the system quantities.

Entity	Scale Factor
Length	λ
Area	$\lambda^{2}$
Volume	$\lambda^{3}$
Mass	$\lambda^{3}$
Velocity	λ
Acceleration	$\lambda^{-1}$
Force	$\lambda^{2}$
Energy	$\lambda^{3}$
Time	λ

**Table 4 materials-14-05884-t004:** Experimental conditions.

	“*A*” Sample	“*B*” Sample
length *a* (mm)	50	100
length *b* (mm)	100	200
Laminate thickness *h* (mm)	1.5	3
Lay thickness *hp* (mm)	1.25	1.25
Number of plies	12	24
Lamination sequence	${\left[0_{2}^{{}^{\circ}}/90_{3}^{{}^{\circ}}/0^{{}^{\circ}}\right]}_{s}$	${\left[0_{4}^{{}^{\circ}}/90_{6}^{{}^{\circ}}/0_{2}^{{}^{\circ}}\right]}_{s}$
Impactor diameter (mm)	10	20
Impactor mass (kg)	1.075	8.6
Velocity impact (ms^−1^)	1.8 and 2.2	1.8 and 2.2
SPC	clamped	clamped
Constraint diameter (mm)	40	40
Constraint distance (mm)	100	200

**Table 5 materials-14-05884-t005:** Parametric analysis for sample A.

	MAT	Form.	*W_max_* (mm)	*F_cmax_* (N)	*F_cfmax_* (N)	*Atc* (ms)	*AU* (mJ)	*EH* (mJ)	*EH_u_* (mJ)
Pinned	MAT22	2	7.8	561.2	435.64	13.5	645.6	88.1	214.2
		8	7.6	603.0	484.14	12.9	25.0	4.3	-
		10	7.6	523.8	480.60	13.4	16.0	31.1	-
		16	7.7	586.4	473.43	13.4	102.9	-	-
	MAT54	2	7.5	614.5	436.14	13.0	817.1	45.7	191.4
		8	7.6	617.3	484.98	12.9	25.1	3.8	-
		10	7.6	524.9	480.65	13.3	12.0	33.2	-
		16	7.7	617.5	472.07	13.2	81.2	-	-
	MAT58	2	7.8	528.7	462.76	13.3	226.8	113.5	-
		8	7.6	610.2	481.99	13.1	16.6	3.0	-
		10	7.6	532.8	481.89	13.2	12.3	21.1	-
		16	7.7	671.4	459.26	13.2	403.3	-	-
Cylindrical SPC	MAT22	2	7.5	502.7	490.85	14.5	683.3	70.0	358.7
		8	7.2	551.6	536.69	12.5	18.3	4.0	-
		10	7.2	535.5	528.25	12,7	24.8	15.8	343.6
		16	7.3	550.6	529.65	12.7	16.1	-	-
	MAT54	2	7.5	520.5	511.39	12.4	305.1	54.5	343.2
		8	7.2	553.2	537.22	12.4	8.9	4.2	-
		10	7.2	545.6	532.22	12.5	11.3	18.2	344.4
		16	7.3	552.8	529.88	12.7	15.2	-	-
	MAT58	2	7.1	514.1	484.16	13.5	157.0	21.4	195.4
		8	7.2	559.1	537.22	12.5	11.9	3.3	-
		10	6.9	534.0	520.30	12.7	65.6	6.4	249.9
		16	7.3	552.2	530.33	12.7	22.6	-	-

**Table 6 materials-14-05884-t006:** Parametric analysis for sample B.

	MAT	Form.	*W_max_* (mm)	*F_cmax_* (N)	*F_cfmax_* (N)	*Atc* (ms)	*AU* (mJ)	*EH* (mJ)	*EH_u_* (mJ)
Pinned	MAT22	2	16.0	2359.8	1796.3	27.4	5866.0	1147.0	2282.3
		8	15.4	2614.1	1985.1	26.6	180.1	29.4	-
		10	15.5	2158.1	1971.0	26.5	113.2	265.0	-
		16	15.7	2577.2	1938.3	26.3	1572.7	-	-
	MAT54	2	15.9	2556.9	1914.1	27.1	3408.0	493.2	1700.2
		8	15.4	2645.7	1988.6	26.6	74.7	32.7	-
		10	15.5	2153.0	1973.2	26.9	108.6	257.4	-
		16	15.7	2746.1	1939.9	26.0	1778.6	-	-
	MAT58	2	15.9	2176.9	1899.8	27.1	1855.8	905.8	-
		8	15.4	2587.2	1986.3	26.9	105.8	26.0	-
		10	15.5	2185.5	1976.4	26.8	67.1	183.6	-
		16	15.6	2810.5	1833.1	24.6	4823.9	-	-
Cylindrical SPC	MAT22	2	15.7	2003.5	1853.8	28,.2	4831.8	637.9	3708.8
		8	15.1	2159.2	2023.0	25.9	85.3	25.8	-
		10	15.1	2134.7	2018.1	26.3	574.1	104.5	6480.4
		16	15.3	2111.8	1996.3	26.4	187.1	-	-
	MAT54	2	15.6	2097.5	1952.3	26.0	4153.5	604.5	2232.5
		8	15.1	2178.2	2026.8	25.9	51.4	27.8	-
		10	15.1	2155.4	2021.7	26.0	158.5	97.5	6509.4
		16	15.3	2124.4	1996.8	26.6	329.1	-	-
	MAT58	2	15.4	2040.5	1999.5	26.1	1460.3	1320.2	-
		8	15.1	2133.3	2022.9	26.1	65.5	23.4	-
		10	14.5	2265.8	1955.7	26.4	608.2	96.3	6900.8
		16	15.9	2232.8	1947.0	26.8	6988.4	-	-

**Table 7 materials-14-05884-t007:** Indices for the choice of the numerical setup. Comparison with the evidence.

*k*	$V_{W}^{kT}$	$V_{F}^{kT}$	$V_{\Delta t_{c}}^{kT}$	$V_{\Delta U}^{kT}$	$V_{Hmax}^{kT}$	*ID^k^*
**SFSF**	3.0%	11.3%	9.2%	10.4%	1.8%	7.1%
**CS**	−1.0%	−0.3%	7.1%	8.6%	1.4%	3.7%
**MAT 22**	1.4%	3.9%	9.5%	10.7%	1.8%	5.5%
**MAT 54**	1.1%	6.9%	7.2%	8.1%	1.3%	4.9%
**MAT 58**	0.4%	5.6%	7.7%	9.7%	1.8%	5.0%
**form 2**	2.7%	1.5%	10.8%	24.5%	4.9%	8.9%
**form 8**	0.0%	9.6%	6.8%	0.6%	0.2%	3.4%
**form 10**	−0.3%	0.1%	8.1%	1.4%	1.2%	2.2%
**form 16**	1.5%	10.9%	6.9%	11.5%	0.0%	6.2%

**Table 8 materials-14-05884-t008:** Indices for the choice of the numerical setup. Comparison with the evidence.

*k*	$V_{W}^{kBA}$	$V_{F}^{kBA}$	$V_{F_{f}}^{kBA}$	$V_{\Delta t_{c}}^{kBA}$	*ID^kBA^*
**SFSF**	2.5%	5.6%	3.2%	0.7%	3.0%
**CS**	−4.9%	−1.3%	−4.8%	2.5%	3.4%
**MAT 22**	3.5%	2.4%	−1.39%	1.2%	2.1%
**MAT 54**	3.9%	2.4%	−0.4%	3.0%	2.4%
**MAT 58**	3.7%	1.7%	−0.5%	0.5%	1.6%
**form 2**	4.7%	2.0%	1.4%	1.0%	2.3%
**form 8**	3.4%	2.2%	−1.6%	3.4%	2.6%
**form 10**	3.3%	2.1%	−1.3%	2.2%	2.2%
**form 16**	3.4%	2.4%	−1.7%	-0.3%	1.9%

**Table 9 materials-14-05884-t009:** Identification of the minimum kinetic energy of impact for failure.

*v_i_* (m/s)	Sample “*A*”	Sample “*B*”
1.8	elastic	elastic
1.9	elastic	elastic
2.0	elastic	elastic
2.1	elastic	elastic
2.2	elastic	failure
2.3	elastic	failure
2.4	failure	failure
2.5	failure	failure

## Data Availability

Not applicable.

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
