# Peer review of "Validity and Applicability of the Scaling Effects for Low Velocity Impact on Composite Plates"

_materials, 2021, doi:10.3390/ma14195884_

Round 1

Reviewer 1 Report

This paper presents small but interesting study of application of the "Theory of Dimensions and Similarity" to deformation and breakage of composite structures under impact. The text is easy-to-read, though too long sometimes, so it can be shortened. Furthermore:

  1. The Introduction contains brief history of the previous studies, but the ultimate engineering application seems not evident from it. It fact, this paper causes a wish to check its results in a shooting gallery, because most popular applications (defense from weapons in military engineering and from meteorites in aerospace one) operates with much larger impact velocities.
  2. Application of \pi-theorem in Section 2 seems too long, trivial and can be shortened a little. By the way, the author uses three forms of the name of this theorem: "Buckingham Theorem",  "Buckingham Pi-Theorem" and " "Buckingham \pi-Theorem" (with Greek symbol). It can be unified, and I would prefer the last version, because the family of Buckingham is omitted in some countries, so the text becomes unclear for some readers. 
  3. Conditions of experiment that the author obtained seem unclear. It is useful to write about the sensor used, their positions in experiments, accuracy and inertia features.
  4. Some typos are present, as I can understand, for example, "equation (f)" in Page 6, or it has some other sense?
  5. The authors certainly can remain the Conclusions as there are. But can we write for clarity that the chosen scaling model works adequately within the limits of elastic and plastic deformation, and it ceases to work when destruction starts? At this moment, I understand the basic conclusion from this study namely so.

Author Response

Thank you very much for the time and attention given in the comments that the reviewer was able to report. I fixed everything by reporting a comment in a different color in the paper (I chose red), I am convinced that these comments have improved the general readability of the paper. Thanks!

This paper presents small but interesting study of application of the "Theory of Dimensions and Similarity" to deformation and breakage of composite structures under impact. The text is easy-to-read, though too long sometimes, so it can be shortened. Furthermore:

  1. The Introduction contains brief history of the previous studies, but the ultimate engineering application seems not evident from it. It fact, this paper causes a wish to check its results in a shooting gallery, because most popular applications (defense from weapons in military engineering and from meteorites in aerospace one) operates with much larger impact velocities.

I modified the introduction section following your comment, this part is improved.

  1. Application of \pi-theorem in Section 2 seems too long, trivial and can be shortened a little. By the way, the author uses three forms of the name of this theorem: "Buckingham Theorem", "Buckingham Pi-Theorem" and " "Buckingham \pi-Theorem" (with Greek symbol). It can be unified, and I would prefer the last version, because the family of Buckingham is omitted in some countries, so the text becomes unclear for some readers. 

Ok, I integrated it, I made uniform the section 2, but thanks for your comment, I preferred to rewrite this section.

  1. Conditions of experiment that the author obtained seem unclear. It is useful to write about the sensor used, their positions in experiments, accuracy and inertia features.

The paper reports the results of experimental tests present in the bibliography on samples of different scales subject to low velocity impacts at different energy levels. The outputs of these results are considered to our investigation.

  1. Some typos are present, as I can understand, for example, "equation (f)" in Page 6, or it has some other sense?

Ok, I modified it

  1. The authors certainly can remain the Conclusions as there are. But can we write for clarity that the chosen scaling model works adequately within the limits of elastic and plastic deformation, and it ceases to work when destruction starts? At this moment, I understand the basic conclusion from this study namely so.

Thanks for your comment, after this I preferred to rewrite the conclusion section.

Reviewer 2 Report

Some elements should be improved:

- The abstract should be rewritten. The aim, methods, results, in a nutshell, is necessary.

- The introduction is too long. It should clearly indicate the problem, lacks in the current knowledge, similar works which are close to resolving the problem. After that, the novelty of your solution should be described.

- in the results section, the discussion is totally missed. Please add discussion with the literature data. Without that, it is impossible to make a confrontation of the results with good known knowledge. I do not have a problem with the model and results. However, comparison and explanations in terms of other scientists are necessary.

- the conclusions are too general. The expressions like "cannot, can lead..." are not appropriate. Please rewrite the conclusions, they should describe exactly your achievements. Some specific information should be demonstrated. Some universal knowledge also should be presented. The best way is to add information "why" in the places where you used "can/cannot"

Author Response

Thank you very much for the time and attention given in the comments that the reviewer was able to report. I fixed everything by reporting a comment in a different color in the paper (I chose red), I am convinced that these comments have improved the general readability of the paper. Thanks!

  1. The abstract should be rewritten. The aim, methods, results, in a nutshell, is necessary.

Thanks for your comment, after this I preferred to rewrite the abstract section.

  1. The introduction is too long. It should clearly indicate the problem, lacks in the current knowledge, similar works which are close to resolving the problem. After that, the novelty of your solution should be described.

I modified the introduction section following your comment, this part is improved.

  1. in the results section, the discussion is totally missed. Please add discussion with the literature data. Without that, it is impossible to make a confrontation of the results with good known knowledge. I do not have a problem with the model and results. However, comparison and explanations in terms of other scientists are necessary.

Thanks for your comment, so I preferred to rewrite this section.

  1. the conclusions are too general. The expressions like "cannot, can lead..." are not appropriate. Please rewrite the conclusions, they should describe exactly your achievements. Some specific information should be demonstrated. Some universal knowledge also should be presented. The best way is to add information "why" in the places where you used "can/cannot"

Thanks for your comment, after this I preferred to rewrite the conclusion section.

Reviewer 3 Report

The paper presents a dimensional analysis of impact deformation of composite plates. The dimensional analysis taken from beam geometry is used to scale the loadline displacement and good agreement is reached for scaling data with no specimen damage. For specimen damage, the impactor velocity necessary to start damage has to be scaled by additional factor that in turn depends on actual gemometrical scale factor. To achieve the goal FEM computation is perfomed and verified against experiment, then the correction factor is computed for single geometricals caling.

The article is very complex, and many aspects would be extremely difficult to understand by general materials audience. Moreover its results seems to be very general, i.e. that scaling of impact intensities depends on a geometrical scaling by an arbitrary function evaluated by complex FEM simulation in single point. Despite the perfomed simulations and dimensional analysis are complex, but represent known and well established approaches and thus bring limited novelty to tha paper. Therefore I recommend to REJECT the paper.

Author Response

Thank you very much for the time and attention given in the comments that the reviewer was able to report. I fixed everything by reporting a comment in a different color in the paper (I chose red), I am convinced that these comments have improved the general readability of the paper. Thanks!

The paper presents a dimensional analysis of impact deformation of composite plates. The dimensional analysis taken from beam geometry is used to scale the loadline displacement and good agreement is reached for scaling data with no specimen damage. For specimen damage, the impactor velocity necessary to start damage has to be scaled by additional factor that in turn depends on actual gemometrical scale factor. To achieve the goal FEM computation is perfomed and verified against experiment, then the correction factor is computed for single geometricals caling.

The article is very complex, and many aspects would be extremely difficult to understand by general materials audience. Moreover its results seems to be very general, i.e. that scaling of impact intensities depends on a geometrical scaling by an arbitrary function evaluated by complex FEM simulation in single point. Despite the perfomed simulations and dimensional analysis are complex, but represent known and well established approaches and thus bring limited novelty to tha paper. Therefore I recommend to REJECT the paper.

The article has been completely revised thanks to the comments of 3 reviewers who have engaged in their work, an in-depth work has been carried out on the different sections, your comments and your perplexities are certainly included in this second issue of the paper. Let me know please if thbis second issue satisfied your judgement.

Round 2

Reviewer 2 Report

The corrections are satisfied.
I recommend publishing the paper.